# The Clinical Potential of Circulating miRNAs as Biomarkers: Present and Future Applications for Diagnosis and Prognosis of Age-Associated Bone Diseases

**DOI:** 10.3390/biom10040589

**Published:** 2020-04-11

**Authors:** Michela Bottani, Giuseppe Banfi, Giovanni Lombardi

**Affiliations:** 1Laboratory of Experimental Biochemistry and Molecular Biology, IRCCS Istituto Ortopedico Galeazzi, Via Riccardo Galeazzi 4, 20161 Milano, Italy; michela.bottani@grupposandonato.it (M.B.); banfi.giuseppe@fondazionesanraffaele.it (G.B.); 2Vita-Salute San Raffaele University, 20132 Milano, Italy; 3Department of Athletics, Strength and Conditioning, Poznań University of Physical Education, Królowej Jadwigi 27/39, 61-871 Poznań, Poland

**Keywords:** biomarkers, circulating miRNAs, miRNA signature, extra-analytical variability, sensitivity and specificity, osteoporosis, fracture risk, osteoarthritis

## Abstract

Osteoporosis, related fracture/fragility, and osteoarthritis are age-related pathologies that, over recent years, have seen increasing incidence and prevalence due to population ageing. The diagnostic approaches to these pathologies suffer from limited sensitivity and specificity, also in monitoring the disease progression or treatment. For this reason, new biomarkers are desirable for improving the management of osteoporosis and osteoarthritis patients. The non-coding RNAs, called miRNAs, are key post-transcriptional factors in bone homeostasis, and promising circulating biomarkers for pathological conditions in which to perform a biopsy can be problematic. In fact, miRNAs can easily be detected in biological fluids (i.e., blood, serum, plasma) using methods with elevated sensitivity and specificity (RT-qPCR, microarray, and NGS). However, the analytical phases required for miRNAs’ evaluation still present some practical issues that limit their use in clinical practice. This review reveals miRNAs’ potential as circulating biomarkers for evaluating predisposition, diagnosis, and prognosis of osteoporosis (postmenopausal or idiopathic), bone fracture/fragility, and osteoarthritis, with a focus on pre-analytical, analytical, and post-analytical protocols used for their validation and thus on their clinical applicability. These evidences may support the definition of early diagnostic tools based on circulating miRNAs for bone diseases and osteoarthritis as well as for monitoring the effects of specific treatments.

## 1. Introduction

### 1.1. Biogenesis of miRNAs

In the last years, a great effort has been made in searching novel circulating biomarkers that are able to identify pathological conditions and to support medical decisions. Among these, microRNAs (miRNAs), defined as non-coding RNA molecules of ~22 nucleotides and first discovered in 1993 [1], have been found to exert fundamental modulatory roles in gene expression of almost any human tissues as well as viruses and in all the other living kingdoms [2,3,4,5,6,7]. miRNAs expression can be tissue-/cell- specific as well as ubiquitous. The miRNAs expression profile has been characterized in 61 tissues [8] and their presence has been discovered in 12 human fluids [9], including plasma and serum, thus supporting their biomarker potential [10]. This led to the creations of miRNAs databases in which all the information about every discovered miRNA is described. Currently, according to the most recent update (October 2018) of miRBase (www.mirbase.org), ~2600 human mature miRNAs have been identified [11]. This database also contains information about universal miRNA nomenclature, which is well schematized in [12]. miRNA coding genes have been found in each human chromosome, including the Y and X one, and can be situated in protein-coding gene sequences (introns or untranslated regions) as well as in non-coding gene sequences [13,14,15,16,17].

miRNAs biogenesis, which begins with the formation of a long primary transcript named pri-miRNA, includes numerous steps of maturation. First, pri-miRNA is processed in the nucleus by the DROSHA/DGCR8 complex to obtain the hairpin pre-miRNA, which undergoes to a cleavage in the cytoplasm by DICER that leads to the formation of a double stranded miRNA. Both the strands can be incorporated into the RISC (RNA induced silencing complex) as mature miRNAs for targeting and binding the paired mRNA at the level of the “seed” miRNA region composed by the nucleotides in positions 2–7. Parallel to this canonical miRNA biogenesis, other pathways independent of DROSHA or DICER have been discovered [16,17]. miRNAs expression may be regulated either by epigenetic modifications of the DNA sequences coding for miRNAs (histone modifications and DNA methylation) [18,19,20,21] or by an altered activity or expression of the proteins implicated in miRNA biogenesis [22].

The fundamental biological role of miRNAs, and consequently of the machinery involved in their biogenesis, has been demonstrated in mice in terms of embryonic lethality [23,24] and in mouse and human embryonic stem cells in terms of compromised proliferation and differentiation [25,26].

### 1.2. miRNAs as Biomarkers

The assessment of novel biomarkers for the clinical practice requires standardized pre-analytical, analytical, and post-analytical protocols aimed at providing precise, reproducible and consistent measurements at low cost and in a reasonable time, through the control of all the variables that can introduce biases in the biomarker detection (type of sample, its manipulation, the method for its measurement) [27]. The pre-analytical phase contemplates the management of all the subject-related factors (age, gender, lifestyle habits, acute and chronic diseases, medications) that may affect the concentration/expression level of the analyte, and sample-associated variables (phlebotomy procedure, choice of matrix, sample processing procedure and delay, sample storing delay and conditions) that may affect its measurement. The analytical phase, instead, refers to the method/platform of measurement. Finally, the post-analytical phase is related to all the procedure that bring to the association of the analytical output to the subject from whom the sample has been taken. In the case of miRNA analysis, the analytical output is represented by a raw value (e.g., the threshold cycle [C_T_] from real-time PCR) that needs to be normalized throughout the application of specific, although not universally accepted, algorithms [12,28,29]. Further information about the pre-analytical, analytical and post-analytical variables affecting miRNA determination are reported in Section 1.2.2.

In addition, the novel biomarker must have a significant correlation with the considered pathology or its prognosis. Moreover, it must give information about the considered disease, or that a specific therapy can reduce the related risk, with more elevated performances than the existing methodologies, and having a consistent impact on patient management [27]. Unfortunately, the miRNAs validation process still presents various issues to be solved, but the use of circulating miRNAs as biomarkers for clinical practice will soon be applied and this is particularly evident when miRNAs are used as diagnostic and/or prognostic biomarkers in tumors [30].

#### 1.2.1. Strengths

The use of circulating miRNAs as biomarkers in the clinical practice and research show several advantageous features: (i) they can be evaluated in human biofluids (e.g., urine, plasma, and serum) obtained with minimal/non-invasive procedures [9,31]; (ii) their detection is based on polymerase chain reaction (PCR) methods, especially the real time quantitative PCR (RT-qPCR) that gives the most reproducible results with elevated specificity and sensitivity [32]; (iii) in human biofluids, they can be found both to bind to proteins or to be enclosed in exosomes, ectosomes or in high-density lipoproteins [33,34,35], thus showing an elevated stability.

miRNAs concentration, in plasma samples, measured by qPCR, revealed an elevated variability [9,36], that is physiological for healthy populations, and the levels of circulating miRNAs can be affected by any variables of blood processing [37,38,39].

#### 1.2.2. Weakness

Considering the pre-analytical phase of the miRNA validation process, it can be affected by several biases regarding both patients and samples features.

• Patients

In patients, the circulating miRNA levels can be modified by: (i) the circadian rhythm [40]; (ii) the dietary regime [41]; (iii) the physical activity done [42]; (iv) smoking habits [43]; (v) the presence of kidney pathologies [44]. Therefore, the patient exclusion and inclusion criteria for the pre-analytical phase must consider all the above-described variables to reduce the errors that can negatively affect the validation of a novel biomarker [31].

The physical activity status is an important determinant of the baseline level of circulating miRNAs. According to our recently published article [45], the physical activity status is responsible for the differences among sedentary and resting trained subjects (mountain marathoners) in skeletal muscle and bone function associated circulating miRNAs according to [46,47,48], but also in cancer-associated circulating miRNAs according to [49].

• Source/matrix, sample collection and handling

Other fundamental aspects for the biomarker validation process are the identification of the proper matrix/source [50,51], the collection, and the manipulation of the chosen sample. In case of healthy subjects, the quantification of miRNA levels both in plasma and serum samples shows reliable, stable, and reproducible results [10,38]. Considering the blood collection process, miRNAs determination can be affected by the needle gauge [39], the tourniquet employment and its prolonged maintenance, free hemoglobin, muscle derived enzymes, molecules with low molecular weight, electrolytes, and water concentration [52,53]. In the collected sample, quantification of circulating miRNAs can be altered by non-circulating miRNAs derived from blood cells, erythrocytes disruption, the piece of skin resulting from the venipuncture site, and platelet activation [37,39,54,55]. Another important source of variability is the type of anticoagulant additive (sodium citrate, lithium or sodium heparin, sodium fluoride/potassium oxalate (NaF/KOx), and ethylendiaminotretracetic acid potassium salts (K2/K3 EDTA)) and collection tube used. In case of miRNAs determination in plasma samples, EDTA is considered the best choice because it does not alter the enzyme activity used in the PCR-based miRNA evaluation platforms and can be removed from the mastermix [56], while heparin and sodium citrate can interfere [31,56,57]. Despite increasing miRNAs detection levels, NaF/KOx can be used as an alternative to EDTA [58]. We have recently demonstrated that detectability and stability of circulating miRNAs in K2EDTA tubes, over different temperatures and delays of storage, is improved by the presence of a gel separator (PPT™) compared to standard tubes. Moreover, K2EDTA samples undergone to platelet depletion (double centrifugation) both detectability and stability were significantly worsened [29].

In plasma, an additional source of error in miRNAs isolation can derive from centrifugation settings (time and force) that can cause platelets’ miRNAs release [37]. Interestingly, miRNA levels in blood, serum, and plasma samples remain unchanged up to 1 day after collection, when kept at room temperature [38], or for decades, when stored at temperatures lower than -70 °C [56]. It is also known that miRNAs are stable at extreme pH values and after recurrent cycles of sample thawing and freezing [10].

Considering all the above-described pre-analytical variables that can affect circulating miRNAs quantification, it is fundamental to standardize both sample collection and handling protocols to reduce the bias that can affect the pre-analytical phase of miRNAs validation [37].

Considering the analytical phase, an important source of error for miRNAs evaluation derive from the quantification platform used. The majority of the papers, aimed at validating miRNAs as novel circulating biomarkers for a specific disease, have employed different quantification platforms (next generation sequencing, microarray, and PCR) for screening and validation stage, but results obtained from different technologies show significant variances in evaluating miRNAs levels [59]. Thus, the analytical protocols and the platform used must be unchanged [31].

In the post-analytical stage, the reference gene selection together with the normalization strategy applied for miRNAs quantification represent the most challenging issue due to the absence of a standardized methodology. The RT-qPCR data for miRNAs expression can be normalized using single or multiple endogenous or exogenous reference genes as well as by the averaged expression value of all the measured miRNAs [28,38,60,61,62,63]. Advantages and disadvantages of these normalization strategies are well summarized in [28], which also highlights how the best normalization strategy consists of the use of the reference gene with the most stable expression values in all the samples of a specific trial [28].

Considering all the pre-analytical, analytical, and post-analytical variables that can affect miRNAs quantification and validation, it is fundamental to draw detailed and standardized guidelines for obtaining consistent and comparable data of miRNAs expression in different studies and labs. This process is necessary to obtain reliable results that lead to the introduction of circulating miRNAs as biomarkers in clinical practice.

## 2. miRNAs as Circulating Biomarkers in Bone Diseases

Given their huge potentialities, the interest around the use of miRNAs as biomarkers has enormously increased, as testified by the steep increase in the number of published papers on this topic in the last few years. This interest regards almost the whole panel of pathological conditions in humans, but it is particularly true in the study of those conditions having somehow limited diagnostic/prognostic tools or with a wide spectrum of clinical presentation. This is the case in bone diseases, and particularly for osteoporosis and osteoarthritis, whose incidence and prevalence are increasing due to the population ageing. These conditions, which are affected by several subject-specific variables, do not have diagnostic tools to predict the subject-specific risk of pathology development/evolution as well as the response to the treatments. For this purpose, circulating miRNAs are promising since they mark epigenetic modifications that regulate and anticipate the possible modification in the downstream classical protein markers [64]. In particular, miRNAs play a fundamental role in bone remodeling especially by regulating of osteoclast and osteoblasts differentiation and function. Thus, altered expression levels of these miRNAs are known to affect osteoblastogenesis and osteoclastogenesis as well as function, apoptosis, and proliferation of bone cells, acting on key regulators of these processes (i.e., Runx2, Wnt signaling pathway, PDCD4, RANKL, c-Fos, Osterix, ALP) [12,65,66]. These alterations lead to an altered bone cell and matrix homeostasis and, as a consequence, to osteoporosis development and progression. Indeed, several papers have identified tissue- and exome/microvesicle-associated miRNAs with key roles in bone physiology and/or pathology. However, the clinical utility of circulating miRNAs in the diagnosis and prognosis of bone and muscle-skeletal diseases, fractures risk or treatment response has not been determined yet. This is probably due to the fact that these papers mainly focused on the mechanistic role of these miRNAs more than on the identification of circulating biomarkers [31]. For this reason, the identification of an altered skeletal metabolism by means of miRNAs as circulating biomarkers is a topic of interest in clinical research.

Based on this growing interest, the aim of this review is to summarize the available data about miRNAs as circulating biomarkers for specific skeletal diseases and their potential use in the clinical practice and research. Information has been extracted from each valuable study focused on circulating miRNAs as potential biomarkers for diagnosis, prognosis, and/or prediction, and are correlated with the information about all the phases (pre-analytical, analytical and post-analytical) required for circulating miRNAs evaluation and validation. These papers were selected from a PubMed literature research using different combination of the keywords: ‘miRNAs’, ‘circulating’, ‘biomarker’, ‘plasma’, ‘serum’, ‘blood’, ‘diagnosis’, ‘prognosis’ and ‘osteoporosis’, ‘fracture’, ‘osteoarthritis’. A certain importance has been given to the information about accuracy, sensitivity and specificity of specific miRNA(s) in assessing the presence, the stage, and the associated risks to give a practical overview. In particular, the plot of sensitivity vs. 1-specifity, called receiver operating characteristic (ROC) curve, outlines the area under the curve (AUC) that, by giving an optimal cut-off, reveals information about the accuracy that the considered circulating miRNA has in discriminating patients from healthy subjects [67]. Beside the different biological roles of free and microvesicle/exosome-associated miRNAs (diagnostic vs. hormone-like) [68], this review has focused on the specific free fraction.

### 2.1. Circulating miRNAs in Pre- and Postmenopausal Osteoporosis

Osteoporosis (OP) is a skeletal disorder consisting of diminished bone quality and strength that lead to an augmented bone fractures risk [69]. Currently, the DXA (Dual energy X-ray Absorptiometry) is the best methodology for OP diagnosis while CTx (C-terminal type I collagen crosslinks), PINP (type I pro-collagen pro-peptide), pyridonline/deoxypyridinoline, PTH (parathyroid hormone), osteocalcin, BAP (bone alkaline phosphatase), and TRAP5b (tartrate-resistant acid phosphatase 5b), which are known markers of bone turnover, are useful in evaluating the bone cells metabolic activity and the efficiency of the anti-resorptive therapies [70,71]. It is well known that these diagnostic methods, although valuable, present several practical defects that limit part of their diagnostic potential: bone turnover markers are not completely accurate in evaluating bone formation and reabsorption, while DXA can give information only about the just established bone architectural alterations which require weeks or also months to be detected [70]. Up to now, miRNAs show a great diagnostic and prognostic potential despite the significant issues in the pre-analytical, analytical, and post-analytical evaluation phases and their ability as modulators of the biological pathways, if evaluated in combination with the commonly used diagnostics, may reveal a more exhaustive clinical situation as well as rapid information of patients’ response to the treatment [28,48]. This is fundamental for OP, which is a complex syndromic disease whose prognosis and evolution (i.e., fracture) depend both on the whole-body metabolism and bone metabolic status. Based on these considerations, circulating miRNAs are more appropriate for describing such a complex network. Hence, the identification of specific miRNAs (miRNA signatures) associated with OP could give additional information, mainly derived from the integration of multiple signals, that complemented with those coming from imaging and biochemical markers would improve the standard diagnostic path and would increase the prognostic potential.

Several studies aimed at validating circulating miRNAs, in serum or plasma as well as in whole blood, as prognostic or diagnostic markers for discriminating OP patients from non-OP subjects. In plasma samples from OP and osteopenia Chinese postmenopausal women, the miR-133a and miR-21 levels are, respectively, increased and decreased compared to healthy subjects and both are associated with BMD [72]. The downregulation of miR-21 has been previously identified in BM-MCSs (bone marrow mesenchymal stem cells) obtained from women with postmenopausal OP [73], whereas low BMD values has been related with the circulating monocytes expression levels of miR-133a [74]. The circulating levels of a miRNA set, consisting of miR-590-5p, miR-194-5p, miR-151a-3p, miR-151b, miR-130b-3p, have been found to be increased in whole blood samples of postmenopausal OP women compared to osteopenia group; among them, the miR-194-5p circulating levels have showed the highest increase and is negatively associated with BMD [75]. Similarly, in women with postmenopausal OP, the miR-27a serum levels are strongly decreased compared to the healthy group [76]. Another potential circulating marker for postmenopausal OP has been identified in plasma miR-148a-3p, which is increased in OP compared to controls. Moreover, plasma miR-126-3p and miR-423-5p levels have revealed a positive correlation with BMD and a negative association with OP fractures 10-year probability, respectively [77]. Starting from the expression levels of 15 miRNAs found differently expressed in a model of OP mice, serum miR-328-3p, miR-142-3p, and miR-103-3p have been confirmed as downregulated in postmenopausal OP women whereas miR-30b-5p levels are significantly decreased in both the OP and osteopenia groups compared to healthy women. These four serum miRNAs show a positive correlation with BMD and their value as diagnostic markers has been revealed by the ROC analysis: AUC of 0.79 (miR-30b-5p) for OP and osteopenia, 0.87 (miR-328-3p), 0.79 (miR-142-3p), and 0.80 (miR-103-3p) for OP compared with healthy group [78]. In a population of Mexican-Mestizo women in post-menopause, the increase of miR-140-3p and miR-23b-3p in OP, osteopenia, and fractured group serum samples, and of miR-885-5p in the osteopenia samples compared with the control group, has been identified. The ROC analysis reveals the potential of miR-140-3p and miR-23b-3p as circulating markers for fracture risk and BMD status: AUC of 0.84 for osteopenia, 0.96 for OP, and 0.92 for fractured compared with healthy subjects (miR-140-3p); 0.73 for osteopenia, 0.69 for OP, and 0.88 for fractured compared with healthy subjects (miR-23b-3p) [79]. Considering the miR-338 cluster, a cohort of postmenopausal OP Chinese women revealed more elevated serum levels of miR-3065-5p and miR-338-3p than the healthy controls, which shown also AUC of 0.87 and 0.74, respectively [80]. Similarly, increased levels of serum miR-483-5p has been found in another study population of OP women vs. non-OP controls [81]. Very recently, an independent study evaluated five bone-specific miRNAs (miR-206, miR-133a-3p, miR-125b-5p, miR-100-5p, and miR-24-3p) in serum samples of postmenopausal OP but results revealed no significant alterations compared to healthy postmenopausal women [82] that, in the case of miR-133a-3p, is in contrast with a previous observation [72]. Circulating miRNAs may be also a potential tool for monitoring the anti-OP therapies effects. Women with postmenopausal OP reveal decreased serum levels of miR-33-3p after 3 months and miR-133a after 12 months of teriparatide treatment [83]. In addition, postmenopausal OP and osteopenia women treated with anti-OP therapy revealed more elevated serum levels of miR-497-5p and miR-181c-5p than the OP and osteopenia postmenopausal women without anti-OP therapy, respectively [84].

Interestingly, one of the latest published works about this topic revealed a panel of three circulating miRNAs able to discriminate both pre- and postmenopausal women compared to their respective non-OP controls. In particular, increased levels of miR-208a-3p together with decreased levels of miR-155-5p has been found to discriminate premenopausal OP from premenopausal non-OP, with AUC, sensitivity, and specificity of 0.82, 77.1%, 82.9% (miR-208a-3p) and 0.90, 94.3%, 77.1% (miR-155-5p); while increased levels of miR-637, miR-208a-3p, and miR-155-5p is able to distinguish postmenopausal OP vs. postmenopausal non-OP with elevated accuracy (AUC, sensitivity, and specificity: 0.81, 77.1%, 85.7% for miR-637; 0.85, 80.0%, 82.9% for miR-208a-3p, and 0.83, 80.0%, 80.0% for miR-155-5p). Moreover, significantly higher levels of miR-637, miR-208a-3p, miR-155-5p have been found in postmenopausal OP vs. premenopausal OP. All the three miRNAs positively correlate with age and negatively with BMD and T-score [85].

Taken together, the above-described evidenced reveal that the circulating miRNA with the highest potential as biomarkers for postmenopausal OP is miR-140-3p (AUC: 0.96 vs. control) [79], while only circulating miR-133a levels have been found both increased in plasma of post-menopausal OP vs. controls [72] and decreased in the serum of OP patients after 12 month of teriparatide treatment [83].

Information about circulating miRNAs related to postmenopausal OP are summarized in Table 1.

### 2.2. Circulating miRNAs, Bone Fragility, and Fracture Risk

In OP subjects, the life quality can be negatively affected by bone fractures and fragility. With this purpose, much work has been done for identifying and validating miRNAs as circulating biomarkers for the OP fractures risk. Currently, the gold standard for the assessment of fragility fracture risk is the combination of BMD and fracture risk assessment tool (FRAX) evaluation [86].

#### 2.2.1. Postmenopausal Osteoporosis

Among 83 miRNAs screened, miR-223-3p, miR-148a-3p, miR-125b-5p, miR-124-3p, miR-122a-5p, miR-100-5p, miR-27a-3p, miR-25-3p, miR-24-3p, miR-23-3p, and miR-21-5p have been found more elevated in serum of OP patients than in non-OP subjects, both with fractures. Of these, 8 miRNAs, together with miR-93, have been confirmed as upregulated in another set of OP fracture sera, compared to non-OP subjects, and the ROC analysis showed the great ability of these circulating miRNAs in differentiating OP from non-OP fracture: 62.5%, 76.4%, 61.4%, 74.1%, 62.9%, 69.0%, 60.3%, 57.4%, and 61.3% of sensitivity; 62.3%, 75.0%, 61.0%, 72.1%, 61.7%, 68.3%, 60.4%, 56.7%, and 61.7% of specificity; AUC of 0.61, 0.76, 0.69, 0.77, 0.69, 0.68, 0.63, 0.63, and 0.63 for miR-148a, miR-125b, miR-124a, miR-122a, miR-100, miR-93, miR-24, miR-23a, and miR-21, respectively [87]. The increased serum levels of miR-148a-3p, miR-125b-5p, miR-124-3p, miR-122-5p, miR-100-5p, miR-93-5p, miR-24-3p, miR-23a-3p, and miR-21-5p were subsequently validated in another independent cohort of fractured postmenopausal OP women [88].

An additional paper revealed that serum miR-125b-5p, miR-122-5p, and miR-21-5p levels are significantly increased in fractured OP women compared with OA women with AUC of 0.76 (miR-125-5p), 0.87 (miR-122-5p), and 0.87 (miR-21-5p) [89]. Similarly, serum miR-328-3p, miR-22-3p, and let-7g-5p levels are significantly decreased in postmenopausal OP fracture women [90].

Recently, serum levels miR-125b, miR-30, and miR-5914 were found to be increased in hip fractured OP women compared to healthy control. Among these, miR-125b levels show the highest increase and a notable diagnostic potential for OP (AUC = 0.898) [91] as revealed also by other 3 papers [87,88,89]. Starting from 14 serum miRNAs just correlated with OP and related fracture, the circulating levels of miR-2861 and miR-124-3p have been found upregulated in postmenopausal women with or without vertebral fractures and reduced bone mass, while serum miR-29a-3p, miR-23a-3p, and miR-21-5p are decreased in comparison to controls. Considering the low bone mass subjects, circulating miR-21-5p is lower in the fractured group (AUC: 0.66, 71% specificity, and 66% sensitivity) [92], which is in contrast with previous observations, also in the case of miR-23-3p [87,88,89]. In serum samples from fractured OP patients miR-145, miR-125b, miR-122a, miR-100, miR-27a-3p, and miR-24-3p levels are higher while miR-144-3p levels are lower than the fractured non-OP group [93]. In postmenopausal OP women with hip fractures, the increased serum levels of miR-133a compared to controls are negatively associated with lumbar spine BMD [94].

Very recently, the comparison of circulating miRNA levels in serum samples of postmenopausal women (divided in healthy control, low BMD and no fractures, low BMD and vertebral fractures but without OP treatment, low BMD and vertebral fractures receiving OP treatment) revealed the following evidences: the increased circulating levels of miR-532-3p, miR-486-3p, miR-375, miR-335-5p, miR-152-3p, miR-143-3p, miR-133b, miR-106b-5p, miR-30e-5p, miR-23a-3p, miR-21-5p, miR-19b-3p in low BMD and vertebral fractures but without OP treatment group and miR-532-3p, miR-375, miR-335-5p, miR-152-3p, miR-133b, miR-127-3p, miR-23a-3p, miR-21-5p, miR-19b-3p in low BMD and vertebral fractures with OP treatment group compared with healthy controls. In addition, the serum levels miR-550a-3p, miR-532-3p, miR-486-3p, miR-375, miR-335-5p, miR-214-3p, miR-152-3p, miR-143-3p, miR-127-3p, miR-106b-5p, miR-30e-5p, miR-23a-3p, miR-21-5p, miR-19b-3p in low BMD and vertebral fractures without OP treatment group and miR-550a-3p, miR-532-3p, miR-486-3p, miR-375, miR-335-5p, miR-214-3p, miR-152-3p, miR-127-3p, miR-30e-5p, miR-23a-3p, miR-21-5p, miR-19b-3p in low BMD and vertebral fractures with OP treatment group are increased when compared with low BMD without the fractures group [95].

Interestingly, the evaluation of 32 miRNAs, previously associated with both OP fractures and bone function, in serum samples from 682 women (a prospective cohort composed by pre and postmenopausal groups with or without prevalent fragility fracture) revealed that none of the selected miRNAs, after age adjustment, were significantly associated with fragility fractures [96].

#### 2.2.2. Secondary Osteoporosis

In a cohort of OP men with femoral neck fracture, the serum levels of miR-148a-3p, miR-124-3p, miR-122-5p, miR-100-5p, miR-93-5p, miR-24-3p, miR-23a-3p, and miR-21-5p have been found to be upregulated compared with the non-OP men control group [88]. In searching for serum miRNA associated with traumatic fractures in either post-menopausal or idiopathic OP (pre-menopausal women and men), miR-335-5p, miR-320a, and miR-152-3p have been found to be increased, whereas miR-550a-3p, miR-532-5p, miR-378a-5p, miR-365a-3p, miR-324-3p, miR-215-5p, miR-186-5p, miR-140-5p, miR-93-5p, miR-30e-5p, miR-29b-3p, miR-19a-3p, miR-19b-3p, miR-16-5p, miR-7-5p, and let-7b-5p have been found to be decreased in both fractured men and women compared to their controls. Of these, miR-550a-3p, miR-335-5p, miR-324-3p, miR-152-3p, miR-140-5p, miR-30e-5p, miR-19a-3p, and miR-19b-3p have showed the highest potential in distinguishing fractured from healthy subjects than markers of bone turnover or BMD (AUC: 0.91, 0.94, 0.95, 0.96, 0.95, 0.96, 0.93, and 0.94, respectively) [97] and miR-550a-3p, miR-324-3p, and miR-29b-3p have been significantly correlated to bone architecture and dynamic histomorphometric parameters but not with bone resorption parameters [98].

In a cohort of 139 patients (both men and women divided in healthy controls, osteopenia and OP patient either with or without fractures), the circulating levels of miR-4516 and miR-122-5p have been found to be decreased in OP patients compared to osteopenia and healthy individuals. Of these, miR-4516 shows the lowest levels in fractured OP, correlates with BMD and reveals a great diagnostic ability for OP: specificity of 62%, sensitivity of 71%, and AUC of 0.73, that increases to 0.75 when combined with miR-122-5p [99]. Similarly, the decreased levels of plasma miR-19b efficiently discriminate OP with (0.95-95.0%-85.4%) or without (0.93-91.3%-80.5%) vertebral fracture from healthy controls [100].

Considering fractures associated with both OP and type 2 diabetes (T2DM), the levels of 375 miRNAs have been screened in serum samples: 23 miRNAs are differently expressed in OP fractures compared to healthy controls, while 48 miRNAs in T2DM fractures compared to healthy subjects. Among them, 18 miRNAs reveal the same altered levels in both OP and T2DM sera. Ten combined four-miRNA models display AUC values in a range of 0.92 to 0.97 for differentiating fracture status in T2DM, and 0.97 to 0.99 in non-T2DM [101].

Taken together, the above-described evidenced reveal that the circulating miRNAs with the highest potential as biomarkers for risk of fracture and bone fragility are miR-550a-3p, miR-335-5p, miR-324-3p, miR-152-3p, miR-140-5p, miR-30e-5p, miR-19a-3p, and miR-19b-3p in the case of both post-menopausal or idiopathic OP [97].

In addition, among the above-described validated miRNAs, circulating miR-125, miR-122, and miR-21 are the most commonly found with altered levels in OP with fracture. The circulating levels of miR-125 have been found to be increased in post-menopausal OP with fractures vs. non-OP fractures [87,88,91,93] and vs. OA fracture [89]. Considering miR-122, higher levels of circulating miR-122a have been revealed in fractured post-menopausal OP vs. fractured non-OP [87,93], while miR-122-5p in fractured post-menopausal OP vs. fractured OA [89] or vs. post-menopausal non-OP [88]. Similarly, miR-21 levels have been found to be increased in post-menopausal OP fractures vs. non-OP fractures [87] as miR-21-5p in fractured post-menopausal OP vs. postmenopausal non-OP [88], vs. fractured OA [89], or vs. healthy controls [95]; while decreased levels of miR-21-5p has been found in post-menopausal vertebral OP fracture vs. non-fractured OP [92]. These miRNAs, although commonly found, are hardly comparable due to the differences in the choice of the patient and control population, the protocols for samples handling, the evaluation platform and the normalization strategies used. For example, recently, Feurer et al. reveal that, after age adjustment, most of the miRNAs previously associated with OP fractures lose their relationship with this condition [96].

Interestingly, the circulating miRNAs signatures identified by [90,97,101] after acute OP fractures, despite differences in the considered populations (post- or pre -menopausal women and men), have allowed the development of a set of 19 miRNAs, named OsteomiR™ panel, able to identify low-traumatic fractures, regardless gender and age. This panel recently revealed to be a cost-effectiveness alternative to DXA, FRAX or no monitoring for fracture risk screening and treatment decisions in an Austrian population of women [102]. Very recently, starting from the OsteomiR™ panel, a pilot study identified also the OsteomiR™ score, based on the combination of 10 miRNA serum levels (miR-582-5p, miR-375, miR-335-5p, miR-320a, miR-188-5p, miR-152-3p, miR-144-5p, miR-141-3p miR-127-3p, and miR-17-5p) in an algorithm, that revealed promising results in the prediction of fragility fracture risk, regardless of age, and might have a more elevated potential than FRAX [103].

Information about circulating miRNAs related to fractured OP are summarized in Table 2.

### 2.3. miRNAs, Fracture Risk and Physical Activity

Physical activity (PA) is a well-established therapy in the prevention of bone-loss associated fracture risk. It is also effective in improving the bone metabolic status and to improve peak bone mass in childhood and adolescence and to limit the age-associated bone loss in late adulthood and older ages [104]. The PA status (e.g., trained vs. sedentary) profoundly modulates miRNAs expression in most of the tissues and organs and, consequently, the profile of miRNAs in circulation [105]. However, the knowledge about the effect of PA on the change in tissue and circulating levels of miRNAs, and especially for those associated with osteoporosis, fracture risk and osteoarthritis, is scarce [105]. The relationship linking bone metabolism and PA is complicated and represents a main reason for the still limited knowledge of the miRNA-regulating role in this condition. Indeed, PA (i) directly affects bone, (ii) affects whole-body metabolism that in turn affects bone, (iii) affects extra-skeletal tissues (e.g., skeletal muscle, immune system, adipose tissue, and nervous system) that release mediators (e.g., myokines, cytokines, adipokines, and neurotransmitters) affecting bone both directly and indirectly, iv) affects the secretion from bone of mediators (such as osteokines) that can modulate the expression, in extra-skeletal tissues, of other mediators targeting bone [106,107]. Recently, we have demonstrated that a protocol of PA (consisting of repeated sprint training in young healthy men for 8-week) modulate miR-148a-3p, miR-125-5p, miR-122-5p, miR-100, miR-93-5p, miR-24-3p, and miR-23a-3p in a more sensitive way than the classical bone metabolic hormones, metabolism markers, and cytokines [48]. Moreover, the physical activity status is responsible for the differences among sedentary and resting trained subjects (mountain marathoners) in skeletal muscle and bone function associated circulating miRNAs (e.g., hsa-miR-335, hsa-miR-148a-3p, hsa-miR-125b-p, has-miR-93-3p, hsa-miR-29a, and hsa-miR-1) but also in cancer-associated circulating miRNAs (e.g., hsa-let7a-5p, hsa-miR-495-3p, hsa-miR-155-5p, hsa-miR-30c-5p, hsa-miR-29c-3p, and hsa-miR-16-2-3p) [45].

## 3. Circulating miRNAs and Osteoarthritis

Osteoarthritis (OA) is the most prevalent form of arthritis and represents an important cause of disability. OA is a degenerative joint disease that leads to a progressive reduction of the articular cartilage and increased synovial inflammation that result in clinical symptoms such as pain and joint rigidity. OA can occur at any joint, but the most common affected ones are knee and hip, often leading to joint replacement [108].

miRNAs exert an important role in cartilage development, for this reason are also key players in the pathology of OA. In particular, altered expression levels of these miRNAs can affect chondrocytes homeostasis by acting on apoptosis, senescence, and autophagy, and can increase the inflammation degree and the cartilage matrix degradation by increasing the production of proinflammatory factors (TNF-α, IL-1, IL-6, COX-2, NO, ROS) and matrix metalloproteinases (MMPs) [109,110].

Currently, X-ray is the standard method for OA diagnosis, but it shows some limitations such as the late stage detectability and difficulties in monitoring the disease progression. Therefore, it is necessary to identify non-invasive and sensitive circulating biomarkers that are able to more accurately diagnose OA onset and severity. As in the case of OP, specific OA-associated miRNA signatures could complement the information derived from imaging and biochemistry.

Plasma miR-132 accurately differentiates OA and rheumatoid arthritis (RA) patients from non-OA/non-RA subjects, with AUC-sensitivity-specificity values of 0.90-83.8%-80.7% for RA and 0.91-84.0%-81.2% for OA [111]. Plasma from another OA patient cohorts reveals increased levels of circulating miR-885-5p, miR-345, miR-195, miR-186, miR-184, miR-146a, miR-126, and miR-93 compared to controls [112]. Interestingly, decreased levels of plasma miR-136 showed a strong ability in discriminating both OA (consisting of K/L grades 2, 3, and 4) from healthy controls (AUC: 0.94) and the different K/L grades [113], while serum let-7e levels inversely correlate with OA severity and the number of knee/hip joint replacement [114]. A microarray profiling has revealed that the levels of circulating miR-486-5p, miR-320b, miR-122-5p, miR-92a-3p, and miR-19b-3p are higher in OA patients compared to control subjects. The combination of miR-486-5p, miR-122-5p, and miR-19b-3p gives the best diagnostic potential (AUC = 0.93, 80.0% sensitivity, 88.0% specificity) with a strong correlation with risk and severity of knee OA [115]. Using the same approach, the reduced levels of serum miR-671-3p, miR-140-3p, and miR-33b-3p identify OA patients and have been associated with OA risk and progression (AUC: 0.87 for miR-671-3p, 0.85 for miR-140-3p, and 0.81 for miR-33b-3p) [116].

Interestingly, in a group of postmenopausal women, increased levels of circulating miR-146a-5p and miR-186-5p associate, respectively, with prevalent knee OA and knee OA incident over the next 4 years [117].

To prevent OA progression and to relieve related symptoms, the most widely used treatments include lifestyle modification, surgery, and pharmaceutical drugs. Currently, celecoxib, a selective COX-2 inhibitor, is a commonly used drug due to its safety and effectiveness in OA patients. A microarray profiling of plasma from OA patients after 6 weeks of treatment has identified miR-320a and miR-126-5p as upregulated, and miR-155-5p and miR-146a-5p as downregulated compared to the pre-treatment profile. Moreover, non-responders experienced post-treatment miR-320a downregulation and miR-146a-5p upregulation whereas the contrary happened in responders [118].

In conclusion, circulating miR-136 and the combination of miR-486-5p, miR-122-5p, and miR-19b-3 have the highest diagnostic potential in distinguishing OA patients from healthy controls [113,115]. Among the above-described validate miRNAs, levels of miR-146a have been found to be increased both in plasma of OA vs. controls [112] and in serum of prevalent knee OA vs. controls [117]; while considering OA who received celecoxib treatment, plasma levels of miR-146a have been found to be increased in non-clinical responders and decreased in clinical responders [118]. Similarly, plasma levels of miR-126 have been found to be increased in OA vs. controls [112] and, after celecoxib treatment, decreased in non-clinical responders and increased in clinical responders [118]. Finally, higher plasma miR-186 levels have been detected both in OA vs. controls [112] and in incident OA over the next 4 years [117].

All the above-described evidence about circulating miRNAs related to OA is reported in Table 3.

## 4. Conclusions

The key modulatory role of miRNAs in multiple biological functions is nowadays a well-established concept, as is their association with several physiological and pathological conditions that is pushing their clinical implementation as biomarkers. The most up-to-date findings in the field of age-related bone and osteoarticular diseases (primary OP, OP fractures, and OA) have found a link, while not a direct association, between the clinical presentation and its prognosis, altered levels of tissue miRNAs and consequently of their circulating counterparts. Although potential markers have been identified, large-scale validations are urgently needed in order to confirm their real applicability. The main issue around the available clinical studies aimed at discovering or identifying circulating miRNAs as markers for these diseases, is that these studies have been built on hardly comparable experimental protocols. Standardization and harmonization of the procedures in this field need much effort. The aim must be the definition of clear and universally accepted guidelines covering all the aspects of miRNAs analysis (pre-analytical, analytical, and post-analytical phases) to be applied from discovery to validation. The identification of more specific circulating biomarkers associated with OP and OA and the risk of developing related complications (e.g., fracture, arthroplasty) is, thus, desirable and miRNA are, thus, optimal candidates since they carry information derived from the integration of multiple signals. However, even the most sensitive miRNA signature loses any significance if used alone and consequently the analysis of their expression/circulating levels can acquire a diagnostic/prognostic significance only if in concert with the information coming from imaging (e.g., DXA, X-ray) and biochemical markers. Only within a complete diagnostic path would the implementation of the analysis of miRNA add value, for instance, by giving deeper insights into the phenotypic feature of the patient (i.e., stratification).

## Figures and Tables

**Table 1 biomolecules-10-00589-t001:** Circulating microRNAs (miRNAs) in pre- and postmenopausal osteoporosis (PM OP).

Study Population	Variables in Pre-Analytical Phase: - Sample matrix - Centrifugation conditions - Storage conditions	Quantification Method - Target miRNAs - Analytical method - Applied normalization strategy		- Validated miRNAs - AUC; sensitivity; specificity	Reference
		Screening	Validation		
PM Chinese women with normal (n = 40), osteopenia (n = 40) or OP (n = 40) range of BMD	**- Plasma** - n.d. - Liquid nitrogen	- miR-146a, miR-133a, and miR-21 - RT-qPCR - miR-16		- ↑ miR-133a and ↓ miR-21 in OP and osteopenia vs. control - n.d.	[72]
Screening: PM Chinese women with OP (n = 25) or with osteopenia (n = 23) Validation: PM Chinese women with OP (n = 32), with osteopenia (n = 30), and with normal range of BMD (n = 24)	**- Whole blood**, treated using RBC Lysis Solution; - 450*g*, 10 min - n.d.	- miR-660-5p, miR-590-5p, miR-194-5p, miR-151a-3p, miR-151b, and miR-130b-3p - RT-qPCR (comprehensive miRNA expression analysis by Agilent Human miRNA microarray) - snRNU6	- miR-194-5p - RT-qPCR - snRNU6	Screening: - ↑ miR-590-5p, miR-194-5p, miR-151b, miR-151a-3p, and miR-130b-3p in OP vs. osteopenia - n.d. Validation: - ↑ miR-194-5p in OP and osteopenia vs. control - n.d.	[75]
Screening: PM Chinese women with OP (n = 5) and healthy premenopausal women (n = 5) Validation: PM Chinese women with OP (n = 81) and healthy premenopausal women (n = 74)	-**Serum** - n.d. - n.d.	- 851 miRNAs - Agilent Human miRNA Microarray - n.d.	- miR-27a - RT-qPCR - snRNU6	- ↓ miR-27a in OP vs. control - n.d.	[76]
PM women divided in controls (n = 57) and OP (n = 17) based on lumbar spine/femoral neck/total hip T-score ≤−2.5 SD	**- EDTA-Plasma** - (i) 10 min, 2800 rpm, 4 °C; (ii) 15 min, 9600*g*, 4 °C - −80 °C	- miR-574-5p, miR-423-5p, miR-199a-3p, miR-148a-3p, miR-126-3p, miR-30d-5p, miR-30e-5p, miR-7d-5p, and miR-7e-5p - RT-qPCR - Combination of miR-16-5p and let-7a-5p		- ↑ miR-148a-3p in OP vs. control - n.d.	[77]
PM women with OP (n = 10), osteopenia (n = 7), or normal (n = 19) range of BMD	**- Serum** - (i) 10 min, 2000*g*, 4 °C; (ii) 20 min, 12,000*g*, 4 °C - −80 °C	- miR-425-5p, miR-361-5p, miR-345-5p, miR-215, miR-191a-3p, miR-185-5p, miR-142-3p, miR-103-3p, miR-30a-5p, miR-30b-5p, miR-30e-5p, miR-29b-3p, and miR-26a-5p - RT-qPCR - miR-25-3p		- ↓ miR-30b-5p in both osteopenia and OP vs. control; ↓ miR-328-3p, miR-142-3p, miR-103-3p in OP vs. HC - 0.87; 80%; 100% (miR-328-3p), 0.79; 70%; 79% (miR-142-3p), 0.80; 80%; 72% (miR-103-3p), 0.79; 70.6%; 79.0% (miR-30b-5p)	[78]
Screening: PM Mexican-Mestizo women with normal (n = 20) and OP (n = 20) hip BMD Validation: PM Mexican-Mestizo women with normal (n = 22), OP (n = 26), and osteopenia (n = 28) hip BMD, and with fractures (n = 21)	**- Serum**, obtained within 1h of collection - n.d. - −80 °C	- 754 miRNAs - TaqMan Array Human MicroRNA A+B Cards Set v3.0 - snRNU6	- miR-885-5p, miR-140-3p, and miR-23b-3p - RT-qPCR - snRNU6	- ↑ miR-885-5p for osteopenia vs. HC; ↑ miR-140-3p and miR-23b-3p for osteopenia, OP, and fracture vs. HC - 0.69 (miR-885-5p), 0.84 for osteopenia, 0.96 for OP, 0.92 for fracture (miR-140-3p), 0.73 for osteopenia, 0.69 for OP, 0.88 for fracture (miR-23b-3p)	[79]
PM Chinese women: HC (n = 15) and OP (n = 15)	**- Serum** - n.d. - n.d.	- miR-3065-5p and miR-338-3p - RT-qPCR - cel-miR-39-3p		- ↑ miR-3065-5p and miR-338-3p in OP vs. HC - 0.87 (miR-3065-5p) 0.74 (miR-338-3p)	[80]
PM OP women (n = 36) and non-OP (n = 36)	**- Serum** - n.d. - n.d.	- miR-483-5p - RT-qPCR - U6		- ↑ miR-483-5p in OP vs. non-OP - n.d.	[81]
PM OP treated with teriparatide (n = 30) or denosumab (n = 30) for 12 months.	**- Serum** - n.d. - −80 °C	- miR-2861, miR-503, miR-422a, miR-335-5p, miR-222-5p, miR-218-5p, miR-135b, miR-133a, miR-124-3p, miR-33-3p, miR-29c-3p, miR-27a, miR-26a-5p, miR-24-2-5p, miR-23a-3p, and miR-21-5p - RT-qPCR - Combination of SNORD95, SNORD96A, RNU6-2		- ↓ miR-33-3p after 3 months of teriparatide treatment; ↓ miR-133a after 12 months of teriparatide treatment - n.d.	[83]
PM women divided into: osteopenia with (n = 26) or without anti-OP therapy (n = 14), OP with (n = 29) or without (n = 17) anti-OP therapy; healthy premenopausal women (HC, n = 14)	**- Serum**, clotted at RT for 1 h - 1000*g*, 10 min - −80 °C	- miR-1290, miR-497-5p, miR-204-3p, miR-181c-5p - RT-qPCR - 5S		- ↓ miR-497-5p, miR-204-3p in osteopenia vs. HC; ↓ miR-497-5p, miR-204-3p, miR-181c-5p in OP vs. HC; ↑ miR-497-5p, miR-181c-5p in both OP and osteopenia with anti-OP therapy vs. OP and osteopenia without anti-OP therapy - n.d.	[84]
PM women divided into: OP (n = 35) and non-OP (n = 35); Premenopausal women divided in: OP (n = 35) and non-OP (n = 35).	**- Serum** - 3000*g*, 15 min - −80 °C	- miR-637, miR-208a-3p, miR-155–5p - RT-qPCR - Hs_Snord68_11		- ↑ miR-208a-3p and ↓ miR-155-5p in premenopausal OP vs. premenopausal non-OP; ↑ miR-637, miR-208a-3p, miR-155-5p in PM OP vs. PM non-OP; ↑ miR-637, miR-208a-3p, miR-155-5p in PM OP vs. premenopausal OP - 0.82;77.1%;82.9% (miR-208a-3p) and 0.90;94.3%;77.1% (miR-155-5p) for premenopausal OP vs. premenopausal non-OP; 0.81;77.1%;85.7% (miR-637), 0.85;80.0%;82.9% (miR-208a-3p), and 0.83;80.0%;80.0% (miR-155-5p) for PM OP vs. PM non-OP;	[85]

n.d.: not determined; AUC: Area Under the Curve; BMD: Bone Mineral Density; HC: Healthy Controls; OP: Osteoporosis; PM: Postmenopausal; RT: Room Temperature; RT-qPCR: Real-time quantitative Polymerase Chain Reaction. ↑: increased levels of the validated circulating miRNA; ↓: decreased levels of the validated circulating miRNA.

**Table 2 biomolecules-10-00589-t002:** Circulating miRNAs in OP fracture.

Study Population	Variables in Pre-Analytical Phase: - Sample matrix - Centrifugation conditions - Storage conditions	Quantification Method - Target miRNAs - Analytical method - Applied normalization strategy		- Validated miRNAs - AUC; sensitivity; specificity	Reference
		Screening	Screening		
Screening: OP (3 men + 7 women) and non-OP (10 women) both with pertrochanteric or femoral neck fracture Validation: OP (30 women) and non-OP (30 women) both with pertrochanteric or femoral neck fracture	**- Serum** - n.d. - n.d.	- 83 miRNAs - Human Serum & Plasma miRNA PCR Array MIHS-106Z - snRNU6 and SNORD96a	- miR-637, miR-223-3p, miR-148a-3p, miR-125b-5p, miR-124-3p, miR-122a-5p, miR-100-5p, miR-93, miR-27a-3p, miR-25-3p, miR-24-3p, miR-23a-3p, and miR-21-5p - RT-qPCR - snRNU6 and SNORD96a	- ↑ miR-148a, miR-125b, miR-124a, miR-122a, miR-100, miR-93, miR-24, miR-23a, and miR-21 in fractured OP vs. fractured non-OP - 0.61; 62.5%; 62.3% (miR-148a), 0.76; 76.4%; 75.0% (miR-125b), 0.69; 61.4%; 61.0% (miR-124a), 0.77; 74.1%; 72.1% (miR-122a), 0.69; 62.9%; 61.7% (miR-100), 0.68; 69.0%; 68.3% (miR-93), 0.63; 60.3%; 60.4% (miR-24), 0.63; 57.4%; 56.7% (miR-23a), 0.63; 61.3%; 61.7% (miR-21)	[87]
OP with femoral neck fracture (7 PM women + 7 men), and non-OP (7 PM women + 7 men)	**- Serum**, left at RT for 30 min - 1900*g*, 10 min - −80 °C	- miR-148a-3p, miR-125b-5p, miR-124-3p, miR-122-5p, miR-100-5p, miR-93-5p, miR-24-3p, miR-23a-3p, and miR-21-5p - RT-qPCR - SNORD96a		- ↑ miR-148a-3p, miR-125b-5p, miR-124-3p, miR-122-5p, miR-100-5p, miR-93-5p, miR-24-3p, miR-23a-3p, and miR-21-5p in PM women with OP fracture vs. non-OP PM women; ↑ miR-148a-3p, miR-124-3p, miR-122-5p, miR-100-5p, miR-93-5p, miR-24-3p, miR-23a-3p, and miR-21-5p in men with OP fracture vs. non-OP men - n.d.	[88]
Screening: Caucasian women with hip OA (n = 5) or with OP subcapital hip fracture (n = 8) Validation: Caucasian women with hip OA (n = 12) or with OP subcapital hip fracture (n = 15)	**- Serum** - n.d. - −80 °C	- 179 miRNAs - The miRCURY LNA Universal RT microRNA PCR, Serum/Plasma Focus microRNA PCR Panel - A logarithmic transformation (log2)	- miR-210, miR-143-3p, miR-125b-5p, miR-122-5p, miR-34a-5p, and miR-21-5p - RT-qPCR - miR-93-5p obtained from the GeNorm algorithm	- ↑ miR-125b-5p, miR-122-5p, and miR-21-5p in fractured OP vs. control - 0.76 (miR-125b-5p), 0.87 (miR-122-5p), 0.87 (miR-21-5p)	[89]
Screening: PM Caucasian women with femoral neck OP fracture (n = 7) and without femoral fractures (n = 7) Validation: PM Caucasian women with femoral neck OP fracture (n = 12) and without femoral fractures (n = 11)	**- Serum**, left for 30 min at RT - 2000*g*, RT, 15 min - −80 °C	- 175 miRNAs - The Exiqon serum/plasma focus panels - The average Cp of the detected miRNAs	- miR-328-3p, miR 133b, miR-22-3p, miR-10a-5p, miR-10b-5p, and let-7g-5p - RT-qPCR - The average Cp of the detected miRNAs	- ↓ miR-328-3p, miR-22-3p, and let-7g-5p in fractured OP vs. non-fractured - n.d.	[90]
PM Chinese women with OP (n = 30) and hip fractures or without OP (n = 30)	**- Serum** - 1500*g* - n.d.	- miR-5914, miR-4665-3p, miR-125b, miR-96, and miR-30 - RT-qPCR - snRNU6		- ↑ miR-5914, miR-125b, and miR-30 in fractured OP vs. controls - 0.70 (miR-5914), 0.90 (miR-125b), 0.76 (miR-30)	[91]
Healthy PM women (n = 30), PM women with low bone mass and vertebral fractures (n = 35) or with low bone mass but without vertebral fractures (n = 35)	**- Serum**, left for 10–60 min at RT - (i) 1900*g*, 4 °C, 10 min; (ii) 16,000*g*, 4 °C, 10 min - −80 °C	- miR-2861, miR-422a, miR-335-5p, miR-218-5p, miR-214-3p, miR-135b-5p, miR-133a-3p, miR-124-3p, miR-33a-5p, miR-29a-3p, miR-26a-5p, miR-24-2-5p, miR-23a-3p, and miR-21-5p - RT-qPCR - SNORD95, SNORD96A, and snRNU6-2		- ↓ miR-21-5p in fractured OP vs. non-fractured OP; ↑ miR-2861, miR-124-3p, and ↓ miR-29a-3p, miR-23a-3p, miR-21-5p in OP vs. controls - 0.66-66%-71% (miR-21-5p) for fractured OP	[92]
OP (n = 45) and non-OP (n = 15) both with femoral fractures	**- Serum** - n.d. - n.d.	- miR-211-5p, miR-145-5p, miR-144-3p, miR-128, miR-125b, miR-122a, miR-100, miR-27a-3p, miR-24-3p, and miR-7-5p - RT-qPCR - snRNU6		-↑ miR-145, miR-122a, miR-125b, miR-100, miR-27a-3p, miR-24-3p and ↓ miR-144-3p in fractured OP vs. fractured non-OP - n.d.	[93]
PM Chinese OP women (n = 10) with hip fractures and HC (n = 10)	**- Serum** - 1500*g* - n.d.	- miR-133a - RT-qPCR - snRNU6		- ↑ miR-133a in fractured OP vs. HC - n.d.	[94]
PM women: HC (n = 42), with low BMD and no fractures (n = 39), with low BMD and vertebral fractures without OP treatment (n = 26), with low BMD and vertebral fractures receiving OP treatment (n = 19)	**- Serum** - n.d. - −80 °C	- miR-550a-3p, miR-532-3p, miR-486-3p, miR-451a, miR-375, miR-335-5p, miR-214-3p, miR-188-5p, miR-152-3p, miR-144-3p, miR-143-3p, miR-133b, miR-127-3p, miR-106b-5p, miR-96-5p, miR-30e-5p, miR-29b-3p, miR-23a-3p, miR-21-5p, miR-19b-3p - RT-qPCR - UniSp4		- ↑ miR-532-3p, miR-486-3p, miR-375, miR-335-5p, miR-152-3p, miR-143-3p, miR-133b, miR-106b-5p, miR-30e-5p, miR-23a-3p, miR-21-5p, miR-19b-3p in low BMD+ vertebral fractures without OP treatment vs. HC; ↑ miR-532-3p, miR-375, miR-335-5p, miR-152-3p, miR-133b, miR-127-3p, miR-23a-3p, miR-21-5p, miR-19b-3p in low BMD+ vertebral fractures+OP treatment vs. HC; ↑ miR-550a-3p, miR-532-3p, miR-486-3p, miR-375, miR-335-5p, miR-214-3p, miR-152-3p, miR-143-3p, miR-127-3p, miR-106b-5p, miR-30e-5p, miR-23a-3p, miR-21-5p, miR-19b-3p in low BMD+ vertebral fractures without OP treatment vs. low BMD; ↑ miR-550a-3p, miR-532-3p, miR-486-3p, miR-375, miR-335-5p, miR-214-3p, miR-152-3p, miR-127-3p, miR-30e-5p, miR-23a-3p, miR-21-5p, miR-19b-3p in low BMD+ vertebral fractures+OP treatment vs. low BMD - n.d.	[95]
PM women (n = 10), premenopausal women (n = 10), and men (n = 16) with PM or idiopathic OP low traumatic fractures; PM women (n = 11), premenopausal women (n = 12), and men (n = 16) without low-traumatic fractures	**- Serum** - n.d. - −80 °C	- 187 miRNAs - RT-qPCR - Global mean		- ↑ miR-335-5p, miR-320a, miR-152-3p and ↓ miR-550a-3p, miR-532-5p, miR-378a-5p, miR-365a-3p, miR-324-3p, miR-186-5p, miR-215-5p, miR-140-5p, miR-93-5p, miR-30e-5p, miR-29b-3p, miR-19a-3p, miR-19b-3p, miR-16-5p, miR-7-5p, and let-7b-5p in fractured groups vs. controls - 0.94 (miR-335-5p), 0.87 (miR-320a), 0.96 (miR-152-3p),0.91 (miR-550a-3p), 0.90 (miR-532-5p), 0.87 (miR-378a-5p), 0.81 (miR-365a-3p), 0.95 (miR-324-3p), 0.90 (miR-186-5p), 0.85 (miR-215-5p), 0.95 (miR-140-5p), 0.88 (miR-93-5p), 0.96 (miR-30e-5p), 0.84 (miR-29b-3p), 0.93 (miR-19a-3p), 0.94 (miR-19b-3p), 0.86 (miR-16-5p), 0.82 (miR-7-5p), 0.85 (let-7b-5p)	[97]
Non-OP (1 man + 11 women), osteopenia without fractures (9 men + 52 women), osteopenia with fractures (2 men + 13 women), OP without fractures (6 men + 27 women), and OP with fractures (2 men + 16 women)	**- Serum or plasma** - (i) 30 min, 2500*g*, RT; (ii) 30 min, 14,000*g*, 4 °C - −80 °C	- 370 miRNAs - Serum and Plasma miRNA PCR arrays - SNORD96A and RNU6-6P	- 40 miRNAs - RT-qPCR - SNORD96A and RNU6-6P	- ↓ plasma miR-4516 and serum miR-122-5p in OP vs. non-OP and osteopenia; ↓ plasma miR-4516 in OP with fracture; - 0.75 (miR-122-5p+miR-4516 panel) for OP, 0.73; 71%; 62% (plasma miR-4516) for OP	[99]
Screening: HC (4 women + 2 men), OP (5 women + 1 man) without vertebral fracture, OP (6 women) with vertebral fracture Validation: HC (14 women + 10 men), OP (23 women + 1 man) without vertebral fracture, OP (23 women + 1 man) with vertebral fracture	**- Plasma** - n.d. - n.d.	- 384 miRNAs - Microarray - n.d.	- miR-19b - RT-qPCR - snRNU6	- ↓ miR-19b in OP with or without vertebral fracture vs. HC - 0.95; 95.0%; 85.4% for fracture 0.93; 91.3%; 80.5% for non-fracture (miR-19b)	[100]
T2DM women with (n = 20) and without (n = 20) fragility fractures since the T2DM onset; non-T2DM PM women with (n = 20) or without (n = 20) OP fragility fracture	**- Serum** - 2000*g*, 15 min - n.d.	- 375 miRNAs - SYBR Green Low-density qPCR platform - Cq values were computed using the second derivative maximum method		- 10 candidate four-miRNA models for T2DM fracture status; 10 candidate four-miRNA models for OP fracture status - AUC in a range of 0.92-0.97 for T2DM fracture; AUC in a range of 0.97-0.99 for OP fracture	[101]

n.d.: not determined; AUC: Area Under the Curve; HC: Healthy Controls; OA: Osteoarthritis; OP: Osteoporosis; PM: Postmenopausal; RT: Room Temperature; RT-qPCR: Real-time quantitative polymerase chain reaction; T2DM: Type 2 diabetes mellitus. ↑ increased levels of the validated circulating miRNA; ↓: decreased levels of the validated circulating miRNA.

**Table 3 biomolecules-10-00589-t003:** Circulating miRNAs related to osteoarthritis (OA).

Study Population	Variables in Pre-Analytical phase: - Sample matrix - Centrifugation conditions - Storage conditions	Quantification Method - Target miRNAs - Analytical method - Applied normalization strategy		- Validated miRNAs - AUC; sensitivity; specificity	Reference
		Screening	Validation		
Knee OA (K/L grade 3, n = 11; K/L grade 4, n = 21), RA (n = 30), and HC (n = 30)	**- EDTA-2K-Plasma** - 400*g*, 7 min - −20 °C	- miR-223, miR-155, miR-146a, miR-132, and miR-16 - RT-qPCR - cel-miR-39		- ↓ miR-132 in RA and OA vs. HC; ↓ miR-16 in OA vs. HC - 0.90; 83.8%; 80.7% for RA and 0.91; 84.0%; 81.2% for OA (miR-132)	[111]
Screening: OA (K/L grade 2, n = 6; K/L grade 3, n = 8) and HC (n = 5) Validation: OA (K/L grade 2, n = 14; K/L grade 3, n = 13) and HC (n = 27)	**- EDTA-Plasma** - (i) 10 min, 1800 *g*, RT; (ii) 5 min, 15,000*g* - −80 °C	- 380 miRNAs - TaqMan Low Density Array ver. 2.0 plate A - MammU6	- miR-885-5p, miR-345, miR-195, miR-186, miR-184, miR-146a, miR-126, and miR-93, miR-30b, miR-29c, miR-20b, and miR-16 - RT-qPCR - MammU6	- ↑ miR-885-5p, miR-345, miR-195, miR-186, miR-184, miR-146a, miR-126, and miR-93 in OA vs. HC - n.d.	[112]
Knee OA (K/L stage 2, n = 22; K/L stage 3, n = 29; K/L stage 4, n = 23) and HC (n = 79)	**- Sodium citrate-Plasma** - n.d. - n.d.	- miR-136 - RT-qPCR - snRNA U6		- ↓ miR-136 in OA K/L grade 2+3+4 vs. HC - ↓ miR-136 in OA K/L grade 4 vs. grade 2 and 3 - ↓ miR-136 in OA K/L grade 3 vs. grade 2 and ↑ vs. grade 4 - ↑ miR-136 in OA K/L grade 2 vs. grade 3 and 4 - 0.94 (miR-136) for OA K/L grade 2+3+4 vs. HC	[113]
Screening: 13 pooled samples for each group of the entire study population [OA with (K/L grade 3/4, n = 67) and without knee/hip arthroplasty (n = 749)] Validation: OA with (K/L grade 3/4, n = 67) and without knee/hip arthroplasty (n = 749)]	**- Serum** - n.d. - n.d.	- 374 miRNAs - Human TaqMan miRNA array (Card A V.2.1) - snRNA U6 and miR-45	- let-7b, let-7e, miR-885-5p, miR-454, miR-342-3p, miR-191, miR-146b, miR-140, miR-122, miR-93, miR-28-3p, and miR-25 - RT-qPCR - Ct average or snRNA U6	- ↓ let-7e in severe OA with knee/hip arthroplasty vs. OA without knee/hip arthroplasty - n.d.	[114]
Screening: 8 pooled samples for each group of the entire study population [Knee OA (n = 100) and matched HC (n = 100)] Validation: Knee OA (n = 100) and matched HC (n = 100)	**- Sodium citrate-Plasma**, left for 3 h at −4 °C - 10 min, 1500*g*, RT - −70 °C	- 2578 miRNAs, 2025 pre-miRNAs - GeneChip miRNA 4.0 Array - The Geoquery package (version 2.34.0) in R language	- miR-1180-3p, miR-887-5p, miR-663a, miR-486-5p, miR-320b, miR-122-5p, miR-92a-3p, and miR-19b-3p - RT-qPCR - snRNA U6	-↑ miR-486-5p (0.89), miR-320b, miR-122-5p (0.63), miR-92a-3p, and miR-19b-3p (0.83) in knee OA vs. HC - 0.93; 80.0%; 88.0% (miR-486-5p+miR-122-5p+miR-19b-3p panel), 0.89 (miR-486-5p), 0.63 (miR-122-5p), 0.83 (miR-19b-3p) for knee OA	[115]
Knee OA (K/L grade 3, n = 4; K/L grade 4, n = 8) and HC (n = 12) undergoing total knee replacement surgery	**- Serum** - n.d. - n.d.	- 2549 miRNAs - Agilent 8×60K miRNA-array platform - Quantile algorithm	- miR-4284, miR-1233-3p, miR-671-3p, miR-663a, miR-150-5p, miR-140-3p, and miR-33b-3p - RT-qPCR - miR-25-1	- ↓ miR-671-3p, miR-140-3p, and miR-33b-3p in OA vs. HC - 0.87 (miR-671-3p), 0.85 (miR-140-3p), 0.81 (miR-33b-3p)	[116]
Screening: PM women with knee OA (K/L grade 2/3, n = 10) and HC (n = 10) Validation: PM women with (i) prevalent knee OA (K/L grade 2/3, n = 43) and age-BMI matched HC (n = 42) (ii) incident knee OA (n = 23) and age-BMI matched HC (n = 25)	**- Serum** - n.d. - n.d.	- miRNA NGS libraries obtained from RNA conversion using NEBNEXT library generation kit - NGS - Values expressed as tags per million	- miR-1299, miR-885-5p, miR-375, miR-345-5p, miR-200a-3p, miR-199a-3p, miR-195-5p, miR-186-5p, miR-184, miR-146a-5p, miR-139-5p, miR-132-3p, miR-126-3p, miR-93-5p miR-29a-3p, miR-29b-3p, miR-29c-3p, miR-16-5p, and let-7e-5p - RT-qPCR - mean of miR-361-5p, miR-222-3p, and miR-191-5p expression levels	-↑ miR-146a-5p in prevalent knee OA vs. HC; ↑ miR-186-5p in incident knee OA over the next 4 years - n.d.	[117]
Screening: knee OA who received celecoxib treatment for 6 weeks (K/L grade 2, n = 4; K/L grade 3, n = 2) Validation: knee OA who received celecoxib treatment for 6 weeks (K/L grade 2, n = 159; K/L grade 3, n = 59)	**- Plasma** - n.d. - n.d.	- 2578 miRNAs and 2025 pre-miRNAs - GeneChip miRNA 4.0 Array - Geoquery package (Version 3.3.3) in R language	- miR-4796, miR-3197, miR-675-3p, miR-320a, miR-210, miR-155-5p, miR-146a-5p, miR-126-5p, miR-92a-3p, and miR-17-3p - RT-qPCR - snRNA U6	-↑ miR-320a, miR-126-5p and ↓ miR-155-5p, miR-146a-5p in OA after 6 w of treatment vs. OA before treatment; ↓ miR-320a and ↑ miR-146a-5p, miR-126-5p in non-clinical responders; ↑ miR-320a, miR-126-5p and ↓ miR-146a-5p in clinical responders - n.d.	[118]

n.d.: not determined; AUC: Area Under the Curve; HC: Healthy Controls; OA: Osteoarthritis; OP: Osteoporosis; RA: Rheumatoid Arthritis; RT: Room Temperature; RT-qPCR: Real-time quantitative polymerase chain reaction. ↑: increased levels of the validated circulating miRNA; ↓: decreased levels of the validated circulating miRNA.

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
