# Peer review of "The Clinical Potential of Circulating miRNAs as Biomarkers: Present and Future Applications for Diagnosis and Prognosis of Age-Associated Bone Diseases"

_biomolecules, 2020, doi:10.3390/biom10040589_

Round 1

Reviewer 1 Report

This manuscript discussed the diagnostic values of miRNA in age-related bone disease (osteoporosis and osteoarthritis). It is very informative but I think it can be improved in several ways:
  1. The authors spent 4 pages illustrating the basis of miRNA and analytical consideration in assaying miRNA. Although informative, I wonder whether it can be written in a more concise manner and move some of the materials to be later section of the manuscript, and related the analytical considerations with the findings of the bone disease studies summarized. 
  2. Can the author briefly describe the process of literature search? 
  3. The authors described the studies one by one in the respective sections. It will be really handy to the readers if the authors can summarize at the end of each section what are the common miRNAs found in all of the studies mentioned in that section. 
  4. I understand that the authors want to focus on the diagnostic values of miRNA, not biological functions. However, it would be useful if the authors can briefly describe the role of miRNA in the pathogenesis of osteoporosis and osteoarthritis at the beginning of the respective section. 
  5. Is there a disease stage-dependent expression of miRNA reported among the patients? For example, patients with grade 4 osteoarthritis will have higher or lower expression of certain miRNA. 
  6. It is important to highlight the potential application of miRNAs in diagnosing osteoporosis and osteoarthritis. Osteoporosis can be diagnosed via DXA easily and osteoarthritis through X-ray although some experience may be needed. Do the miRNA help to stratify the risk of the patients and help to send those at high risk for confirmation test? Can they replace DXA and joint X-ray?

Reviewer 2 Report

The authors have done a nice job reviewing the literature on miRNA and bone diseases. The structure of the review and the flow of material is quite well. The main concern, however, is the literature review is not complete and they have missed some of the important studies in the field. For instance, regarding the part about miRNA panel in osteoporosis, the main contribution to the field is by Hackl M and his team that has led to Osteomir panel. There are also many recent studies linking Osteomir score and that of FRAX. A more thorough literature review is therefore of utmost importance.

Moreover, it would have been interesting for me as a reader if the authors would have concluded which miRNAs have been cited in more articles or had compared them in different ages, populations or genders

Reviewer 3 Report

In this manuscript Bottani et al., reviewed the current situation of miRNAs as prognostic and diagnostic biomarkers for skeletal disorders related to osteoporosis and osteoarthritis. Besides merely listing the miRNAs associated with those bone diseases, the authors have also summarized the advantages/disadvantages of the present pre-analytical, analytical and post-analytical procedures used in the workflow of miRNA discovery. This comprehensive attitude makes this review unique and valuable for the readers. I have only minor suggestions to the authors before the manuscript is accepted for publication.
1. Would be useful to provide clear definition of pre-analytical, analytical and post-analytical measures at the beginning of introduction.
2. Page 2: The about 1900 human miRNAs referred from miRBase is actually the number of annotated precursors, the number of mature miRNAs is over 2600.
3. Please provide an explanation/definition what AUC means when applying ROC analysis in the validation process.

Round 2

Reviewer 1 Report

Thank you for addressing my comments. I don't have further comments. 

Reviewer 2 Report

The authors have completely addressed my concerns and so  the manuscript could be published in its current form